



# Validation of ash optical depth and layer height retrieved from passive satellite sensors using EARLINET and airborne lidar data: The case of the Eyjafjallajökull eruption.

D. Balis[1], M. E. Koukouli[1], N. Siomos[1], S. Dimopoulos[1], L. Mona[2], G. Pappalardo[2], F. Marenco[3], L. Clarisse[4], L. J. Ventress[5], E. Carboni[6], R. G. Grainger[6], P. Wang[7], N. Theys[8] and C. Zehner[9]

[1]*Laboratory of Atmospheric Physics, Aristotle University of Thessaloniki, Greece,*
*Email:balis@auth.gr*

[2]*Consiglio Nazionale delle Ricerche, Istituto di Metodologie per l'Analisi Ambientale (CNR-IMAA), Tito Scalo, Potenza, Italy*

[3]*Met Office, Exeter, United Kingdom*

[4]*Université Libre de Bruxelles, Brussels, Belgium*

[5]*National Centre for Earth Observation, Atmospheric, Oceanic and Planetary Physics, University of Oxford, United Kingdom*

[6]*COMET, Atmospheric, Oceanic and Planetary Physics, University of Oxford, United Kingdom*

[7]*Royal Netherlands Meteorological Institute (KNMI), De Bilt, The Netherlands*

[8]*Belgian Institute for Space Aeronomy (IASB-BIRA), Bruxelles, Belgium*

[9]*European Space Agency, ESRIN, Frascati, Italy*

## Abstract

The vulnerability of the European airspace to volcanic eruptions was brought to the attention of the public and the scientific community by the 2010 eruptions of the Icelandic volcano Eyjafjallajökull. As a consequence of this event ash concentration thresholds replaced the 'zero-tolerance to ash' rule, drastically changing the requirements on satellite ash retrievals. In response to that, ESA funded several projects aiming at creating an optimal *End-to-End System for Volcanic Ash Plume Monitoring and Prediction.* Two of them, namely the SACS-2 and SMASH projects, developed and improved dedicated satellite-derived ash plume and sulphur dioxide level assessments. These estimates were extensively validated using ground-based and aircraft lidar measurements. The validation of volcanic ash levels and height extracted from the GOME-2 and IASI instruments on board the MetOp-A satellite is presented in this work. EARLINET lidar measurements are compared to different satellite retrievals for two eruptive episodes in April and May 2010. Comparisons were made between satellite



retrievals and aircraft lidar data obtained with UK's BAe-146-301 Atmospheric Research
Aircraft (managed by the Facility for Airborne Atmospheric Measurements, FAAM) over the
United Kingdom and the surrounding regions. The validation results are promising for most
satellite products and are within the estimated uncertainties of each of the comparative
datasets, but more collocation scenes are needed to perform a comprehensive statistical
analysis. The satellite estimates and the validation data sets are better correlated for high ash
optical depth values, with correlation coefficients greater than 0.8. The IASI data show a better
consistency concerning the ash optical depth and ash layer height when compared with the
lidar data.

## 1.  INTRODUCTION

The Eyjafjallajökull volcano in Iceland (63.63°N, 19.62°W) erupted on the 14th of April 2010
and the ash-loaded plume rose to more than 10 km, deflected to the east by westerly winds
[Stohl et al., 2011]. The plume persisted from the 15th and the 26th of April 2010, mostly over
central Europe while occasionally extending to southeast Europe [Emeis et al., 2011]. New
significant eruptions occurred between May 4th–9th and May 14th–19th 2010 [Gudmudsson et
al., 2010]. The first of these phases mainly influenced Western Europe, from Great Britain to
the Iberian Peninsula, while the second phase influenced central Europe and the central and
eastern Mediterranean on the May 18th–22nd. Final observations of the event were recorded
over central Europe on the 25th of May [Gudmundsson et al., 2010]. Although the eruption
was a moderate one, due to advection of the volcanic ash plumes, civil aviation was shut down
for many days over numerous European countries [Gertisser., 2010].  This resulted in an
urgent demand for reliable model forecasts of the vertical and horizontal extent of the ash
plume, and for complementary measurements that could be used for nowcasting and forecast
verification [Sears et al., 2013]. Following an eruption, Volcanic Ash Advisory Centres (VAAC)
distributed around the globe give instructions to civil aviation in order avoid potential hazards
[e.g. Guffanti et al., 2010]. Considering the large social and economic impact of any decision,
the provided guidelines should be reliable, verifiable and should use all available scientific
information [Zehner, 2010]. During the eruption period the European Aerosol Research Lidar
Network, EARLINET, responded to this demand with coordinated intensive measurements
from ground-based lidar [e.g. Ansmann et al, 2010; 2011; Groß et al., 2011; Mona et al., 2012;
Papayannis et al., 2012; Perrone et al, 2012; Navas-Guzman et al, 2013; Pappalardo et al,
2013]. This observation campaign provided information on ash height and its vertical extent,



as well as an estimation of the ash load in terms of optical depth and mass concentration. In
addition, there were a number of dedicated airborne campaigns during the eruption that
combined lidar and in-situ measurements of the ash plume [e.g. Marenco et al., 2011;
Schumann et al, 2011]. The volcanic plume was observed from a variety of satellite sensors
such as the Cloud-Aerosol Lidar with Orthogonal Polarization (CALIOP) on board the CALIPSO
satellite [Winker et al., 2012] and a number of passive satellite sensors either in low Earth
orbit, such as GOME-2/MetopA [e.g. Rix et al., 2012], MODIS/Terra & /Aqua [e.g. Christopher
et al., 2012], IASI/MetopA [Carboni et al, 2012], or in geostationary orbit, such as SEVIRI [e.g.
Francis et al., 2012]. The World Meteorological Organization organized an intercomparison
campaign of twenty two satellite-based volcanic ash retrieval algorithms applied on passive
sensors [WMO, 2015]. The intercomparison was based on six selected volcanic eruptions
including Eyjafjallajökull. Validation results showed variable agreement with lidar data,
depending upon the scene conditions.
In 2012 the European Space Agency (ESA) initiated the project "Satellite Monitoring of Ash
and Sulphur Dioxide for the mitigation of Aviation Hazards" (SACS-2) to support authorities
and the VAACs during future volcanic events. The project created an optimal end-to-end
system for volcanic ash plume monitoring and prediction [Brenot et al., 2014 and
http://sacs.aeronomie.be]. The system is based on improved and dedicated satellite-derived
ash plume and sulphur dioxide products, followed by extensive validation using satellite and
ground-based measurements [Koukouli et al., 2014a; Spinetti et al., 2014]. In this paper, we
present validation results for two satellite sensors, GOME-2/MetOp-A and IASI/MetOp-A,
concerning the volcanic ash optical depth and ash layer height, using ground and aircraft lidar
measurements. The comparisons are restricted to the Eyjafjallajökull eruption period of 2010.
In the first section we provide a short description of the satellite data and then a description
of the ground-based and aircraft lidar data used as a reference for validation. Then we
describe the methodology applied in the comparisons, and the co-location criteria applied. In
the second section, we present the comparison results for the different sensors and
algorithms, separately for the ground-based and aircraft data. Finally, we discuss the results
and summarize our findings.
## 2.   DATA AND METHODOLOGY
### 2.1    SATELLITE DATA





One of the main tasks of ESA's SACS-2 and SMASH (*Satellite Monitoring of Ash and Sulphur*
*dioxide for the mitigation of aviation Hazards)* projects was to improve and validate the
algorithms for the retrieval of ash optical depth and height, using satellite measurements in
the infrared and UV-Vis from low Earth orbit sensors. These improvements were based on
previous algorithm developments [e.g. de Graaf et al., 2005; Clerbaux et al., 2009; Clarisse et
al, 2010, 2013; Gangale et al., 2010; Carboni et al., 2012; Grainger et al., 2013] In this paper
we use data from GOME-2 and IASI instruments on board the MetOp-A satellite which covered
the whole eruption period of Eyjafjallajökull in 2010. Details of the satellite data are described
below.

### 2.1.1   GOME-2/METOP-A

Data from GOME-2/MetOp-A have been processed by the Royal Netherlands Meteorological
Institute (KNMI). The volcanic ash retrieval algorithm includes an estimation of the optical
depth of an ash layer based on the Absorbing Aerosol Index (AAI) as well as an estimation of
the effective ash layer height. The algorithm is based on look-up tables formed in terms of the
absorbing aerosol index (AAI), aerosol height, solar zenith angle (SZA), view zenith angle (VZA),
and relative azimuth angle (RAZI). The AAI is sensitive to atmospheric parameters such as
aerosol type, aerosol layer height, and aerosol optical depth (AOD), and to surface height and
scattering geometry [de Graaf et al., 2005]. The most dominant parameters are aerosol optical
thickness and aerosol layer height. In general, thick aerosol layers produce larger AAI values
than thin aerosol layers, while high altitude aerosol layers produce larger AAI values than low
lying aerosol layers [Torres et al., 1998; de Graaf et al., 2005]. If the aerosol type, surface
albedo, and geometries (SZA, VZA, RAZI) are known, aerosol optical thickness can be
calculated using the AAI and aerosol height. The ash layer height is derived using the Fast
REtrieval Scheme for Clouds from Oxygen A-band (FRESCO) algorithm [Wang et al., 2008a]. It
has been demonstrated that FRESCO can retrieve volcanic ash layer height for optically thick
ash plumes [Wang et al., 2012]. The retrieved optical thickness of the ash layer depends on
the assumption of aerosol properties used in the look-up tables (LUTs). The volcanic ash
particles are assumed to be spherical and have a bi-modal log-normal size distribution. Two
different a priori assumptions for the refractive index of strongly absorbing volcanic ash were
tested, indicated later on as DUST and VOLZ [Volz, 1973; Sinyuk et al., 2003.]

### 2.1.2   IASI/METOP-A

Satellite estimates for the ash optical depth and layer height from IASI/Metop-A have been
provided by two institutes, the Université Libre de Bruxelles (ULB) and the University of Oxford





(UOXF). The dataset provided by the ULB was generated by a LUT-based algorithm described
in Moxnes et al. [2014] using two distinct sets of refractive indices: one set provided by Dr.
Dan Peters [private communication] based on recent measurements of Eyjafjallajökull ash,
and the other set using the basaltic ash refractive index data from Pollack et al, 1973 (referred
to as the *Eyja* and *Pollack* datasets respectively). In this paper we show only estimates based
on the Eyja refractive index. For this eruption, the ash plume was assumed to be centred at 5
km and no attempt was made to retrieve ash plume height. The datasets provided by UOXF
assume the *Eyja* refractive index, and both UOXF and ULB algorithms assume a log-normal
particle size distribution with spread 2. The algorithmic processing of UOXF resulted in four
different products: one characterized as the `iterative' algorithm, which provided ash optical
depth and layer height, and three characterized as the `fast' algorithm, which provided ash
optical depth for three fixed volcanic ash layer pressures (400 hPa, 600 hPa and 800 hPa). The
fast algorithm, based on the method of Walker et al. [2011], carries out a linear retrieval (least
squares fit) of the aerosol optical depth, AOD, assuming a fixed plume altitude and effective
radius. The algorithm looks for departures in the measured spectra from an expected
background covariance, created from previous IASI measurements containing no volcanic ash.
The iterative algorithm is a full optimal estimation retrieval using a forward model based on
Radiative Transfer for TOVS, *RTTOV,* a very fast radiative transfer model for nadir-viewing
passive visible, infrared and microwave satellite radiometers. Clear sky radiances from RTTOV
are combined with an ash layer in a method described in detail by Thomas et al. [2009a;
2009b]. RTTOV then provides probable values of AOD, effective radius and plume altitude
[Ventress et al. 2015]. The fast algorithm is used to flag IASI pixels (assuming an AOD threshold
defined by the statistics of the scene) for the presence of volcanic ash, at which point the
iterative retrieval is carried out on the pixel.

## 2.2    LIDAR DATA

The validation of the satellite products used lidar measurements from two sources. The first
was the intensive ground-based lidar measurements from stations that form the European
Research Lidar Network (EARLINET) and the second was the airborne lidar measurements
from the UK's BAe-146-301 Atmospheric Research Aircraft managed by the Facility for
Airborne Atmospheric Measurements (FAAM). The difference between the aircraft Lidar and
the EARLINET stations in absolute AOD loadings may be significant, but is explainable. The
aircraft flights monitored a large area affected by the ash cloud with high ash concentrations.
Meanwhile, for most of the EARLINET stations, the volcanic particles atmospheric content was
almost half of that observed in the UK, which was directly downwind from the eruption.





In the next section we provide a brief description of the lidar measurements used as reference
data for the validation of the satellite products.
**2.2.1   EARLINET DATA**
Lidar data from the EARLINET network [Pappalardo et al., 2014 and http://www.earlinet.org]
were used to validate ash plume height and optical depth. EARLINET was established in 2000
and is the first aerosol lidar network with the main goal of providing data for investigating the
aerosol distribution on a continental scale. EARLINET has established certain protocols for the
measurements and the quality control of the systems and the retrieved data, through
algorithm [Böckmann et al, 2004, Pappalardo et al, 2004] and system [Matthias et al., 2004a,
Freudenthaler et al., 2010, Wandinger et al., 2015), intercomparison campaigns. The network
currently includes 27 stations distributed over the European continent.  The standard
products of EARLINET include aerosol extinction and backscatter profiles. EARLINET data have
been widely used for climatological studies [e.g. Matthias et al., 2004b; Amiridis et al., 2005;
Giannakaki et al., 2007] as well as for monitoring unusual atmospheric events such as desert
dust, biomass burning, pollution episodes, volcanic eruptions and so on. Results have been
presented in numerous publications [e.g. Amiridis et al, 2009; Ansmann et al., 2003 ;
Guerrero-Rascado et al., 2009; Mamouri et al., 2012; Mattis et al., 2010;  Mona et al., 2006 ;
Müller et al., 2007 ; Papayannis et al., 2008; Wang et al., 2008b].
A relational database, containing the output of the 4-D analysis of EARLINET data related to
the volcanic eruption of 2010, has been set up [Mona et al 2012; Pappalardo et al., 2013] and
is freely available on request at http://www.earlinet.org. Information related to the present
study involves aerosol backscatter coefficient profiles for each of the ground-based stations
[The EARLINET publishing group 2000–2010, 2014], as well as a characterization of the
observed layers as pure volcanic or mixed [Pappalardo et al., 2013].  A volcanic aerosol mask
was developed [Mona et al., 2012], which involved aerosol typing, backward trajectory
analyses and model outputs, used together with the lidar measurements at 1 hour temporal
resolution. The data included in the EARLINET database captured the whole Eyjafjallajökull
eruptive event over Europe providing geometrical and optical properties of the tropospheric
volcanic cloud. The volcanic cloud persisted over central Europe for the whole period at
heights of between 3 and 8 km, with maximum load observed on the 16[th] of April 2010 over
Hamburg [Pappalardo et al., 2013]. In our study we only used profiles that corresponded to
pure volcanic, as these were characterized by the methodology applied in Pappalardo et al.,



2013. The list of stations considered for the validation of the satellite products is shown in
**Table I**.
**2.2.2    AIRCRAFT DATA**
The satellite products are validated using lidar measurements from six flights by the UK's BAe-
146-301 Atmospheric Research Aircraft over the United Kingdom and the surrounding seas in
May 2010 [e.g. Marenco et al., 2011; Johnson et al., 2011]. The lidar measurements include
aerosol extinction profiles at 355 nm, which in turn provide plume height and layer optical
depth.  In situ observations were provided by other probes on the aircraft, in particular a
three-wavelength nephelometer and PCASP and CAS optical particle counters; radiative
measurements were taken in the visible and infrared. An example of the available aerosol
extinction profiles, along with flight altitude and flight track is shown in **Figure 1** for the 16[th]
of May 2010. The data shown here will be discussed in more detail in the overview of the
comparison results. In this paper we mainly used lidar data from May 4[th], 5[th], 14[th], 16[th], 17[th]
and 18[th] 2010 flights, when volcanic ash was detected and satellite data were available. Since
the satellite AOD estimates were given at 550 nm we considered scaling the lidar-determined
ash layer optical depth to 550 nm using an appropriate Angstrom exponent. According to
Pappalardo et al. [2013] and based on EARLINET observations, the Angstrom exponent
between 355 and 532 nm ranges between 0.03 and -0.11. So we used an exponent equal to
zero, which practically means that the optical depths to be compared were not scaled.
2.2.2.1    **METHODOLOGY FOR THE EARLINET-SATELLITE COMPARISONS**
The values of each satellite product have been restricted to an area of variable radius around
each EARLINET station, depending on the satellite. The closest point in space and time has
been selected for each overpass, and compared to the respective layer characterized by
EARLINET as volcanic particles. Spatial filtering is applied before the temporal filtering. The
EARLINET relational database for this event contains cases for which two or more volcanic
layers are simultaneously observed in the atmospheric column. For these cases the worst
correlated layer to the satellite data was excluded from analysis. A summary of the satellite
data compared with the EARLINET measurements and the corresponding collocation criteria
can be found in **Table II**.   For all the satellite products a comparison of the AOD has taken
place. For the satellite products that provided volcanic ash layer height information a
comparison of volcanic ash layer height was also performed. The AOD of the EARLINET layers
was derived by the layers' integrated backscatter coefficient multiplied by a fixed extinction-





to-backscatter ratio with a value of 50 sr$^{-1}$ [Ansmann et al., 2010]. We did not use any Raman
lidar measurements since most comparisons were performed for daytime conditions. An
estimated 20% uncertainty on the EARLINET AOD was applied due to the variability of the lidar
ratio for volcanic particles, typically between 40 and 60 sr$^{-1}$ [see Pappalardo et al., 2013 and
references therein]. For the layer height comparison, the height of the centre of mass
provided by the EARLINET database was used, and as estimated layer depth, the distance
between the mass centre from the layer top and base was employed. All the satellite ash
optical depth products were calculated at 550 nm, apart from the KNMI/GOME2 products
which were calculated first at 380 nm and then scaled to 550 nm using appropriate Angstrom
exponents provided by the satellite team. In order to convert the infrared optical depth to
optical depth at 550 nm, both ULB and UOXF teams used the Eyja refractive indices from Dr.
Dan Peters (private communication). Correspondingly, 532 nm lidar measurements were used
in the comparisons.
2.2.2.2    **METHODOLOGY FOR THE AIRCRAFT-SATELLITE COMPARISONS**
The airborne lidar data were available on a per flight basis [Koukouli et al., 2014b] and
included aerosol extinction profiles that provided ash plume height and ash layer optical
depth. The values of these variables were compared with the satellite produced values of ash
optical depth and aerosol layer height (where given) over an area of variable radius ranging
from 50 km to 200 km. The closest satellite value in terms of spatial proximity for every flight
path location was found and presented. Since the overpass times of the satellite data are
around 9:30 L.T. and 21:30 L.T., in order to allow for co-location, only spatial criteria where
used. None of the available aircraft data were available within 1-2 hours of the overpass time,
which was the criterion that provided the best matches when using the EARLINET data. The
time difference between satellite and aircraft data was around 5 hours. This fact does not
allow a point-to-point comparison of the measurements but the comparisons will mainly
highlight whether the ash products from the two measuring systems are consistent. A
summary of the satellite data compared against the flight measurements and the
corresponding collocation criteria can be found in **Table III**.
**3.  RESULTS AND DISCUSSION**
**3.1    COMPARISON OF ASH OPTICAL DEPTH AND ASH LAYER HEIGHT**
**WITH EARLINET DATA**





As shown in **Table *III***, we applied different collocation criteria between the EARLINET lidar
measurements and the satellite observations, to investigate which one provides the best
results and a reasonable number of matches. Although during April and May 2010 the
EARLINET stations performed a large number of dedicated intensive measurements, the
overpass time of the MetOP-A satellite significantly limited the number of collocations. We
examined, for each of the collocation criteria, the correlation coefficient between the lidar-
determined optical depth of the pure volcanic particles layer and the corresponding satellite
estimate. Furthermore, we examined the correlation coefficient between the ash layer height
estimated from the lidar measurements and the one retrieved from the satellite algorithms
when available [Koukouli et al., 2014b]. In Figure 2 we present scatter plots between EARLINET
ash layer optical depth and each satellite ash product for those collocation criteria that
showed the largest correlation. The best correlations were found when limiting the matches
to within a radius of 100 km from the ground-based lidar and considering measurements with
a one-hour difference. When deviating from these criteria, the number of matches increased
but the correlation declined. This fact provides an indication of the spatial and temporal
representativeness of single lidar profiles. Different colours in these plots correspond to
different European regions (see **Table *I***) in order to examine whether the distance from the
source and the transport path have an impact on the comparisons.
The GOME-2A comparisons are shown in the Figure 2a and 2b with the "dust" algorithm in
the left column and the "Volz" algorithm in the right column. Only twelve collocations were
found for the GOME-2 and the EARLINET observations. There is a small correlation between
the datasets, ranging between 0.33 and 0.46 for the "dust" and "Volz" products respectively.
This limited number of co-locations were given by a radius of 300 km from each ground-based
station and within 5 hours. The GOME-2A estimates of the ash layer optical depth are
systematically larger than the lidar ones and most of them are larger than 1, although for
these cases the lidar data rarely exceed the value 0.5. The large GOME-2 pixel size (80 km x
40 km) and the large search radius (300 km) could partly explain differences with point
measurements, like the lidar; however it seems possible that many GOME-2A data are
contaminated by thin clouds, while the lidar data included in the EARLINET database are
always cloud screened. Between the two GOME-2A products the "Volz" algorithm shows a
slightly better correlation coefficient with the ground-based lidars.
The scatter plots of UOXF ash optical depth and collocated EARLINET measurements are
presented in Figure 2c and 2d; the plot in the left column corresponds to the iterative



algorithm and the right column corresponds to the "fast" algorithm at a fixed height of 600
hPa, which is consistent with the average height where EARLINET observed volcanic particles.
For both algorithms the collocation criteria that provided the best results were a distance from
each ground-based station of 100 km and a maximum time difference of one hour. These
criteria allowed for almost 20 coincidences. As it can be quickly verified by the results shown
in Figure 2c and 2d, the ash AOD extracted from the IASI/MetOpA Oxford iterative algorithm
is quite low, with values rarely rising above 0.2, which is consistent with the EARLINET
measurements, which show similar AOD levels. There are only two cases showing AOD values
larger than 0.2 and these are also consistent with EARLINET, since the lidar data for these two
cases show significantly larger values, above 0.4. The correlation coefficient is quite promising
at 0.85, however it is based on only 18 coincident measurements. The agreement between
IASI and EARLINET estimates is similar for the "fast" algorithm, showing a larger scatter for
the low AOD values but potentially less scatter for larger AODs. This larger scatter leads to a
smaller correlation coefficient close to 0.78. If we loosen the collocation criteria to 300 km
and 3 hours then the correlation coefficient drops significantly to a value of less than 0.5.
In Figure 2e we show comparisons of the ash optical depth from the ULB algorithm with
EARLINET estimates. The results are shown for the same collocation criteria applied to UOXF
comparisons, i.e. 100 km distance and one hour difference between the observations. The
general picture is consistent with the IASI/UOXF datasets, however the number of
coincidences decreases. The comparisons show a correlation of 0.91, which is the largest
found in all comparisons shown in Figure 2. **Table *IV*** provides the mean EARLINET and satellite
ash optical depths for the coincidences shown in Figure 2. The average AOD values of the
measurements that meet the collocation criteria are small (less than 0.2) and consistent with
each other, except in the case of GOME-2A and when the IASI-UOXF fast algorithm has a fixed
height of 800 hPa, where the satellite data overestimate the ash optical depth. We have to
repeat that the mean values are based on a small number of coincidences.
The GOME-2A ash products and the iterative IASI product processed by UOXF provided the
height of the ash layer. These heights were compared with the estimates from EARLINET and
the results are shown in **Figure *3***. The ash plume height estimated for GOME-2A products and
the EARLINET network are compared in Figure 3a. Irrespective of the product and the search
radius (not shown here) the comparison is not satisfactory for either of the two algorithms.
The satellite-provided height seems to strongly under-estimate the ground-based values,
showing a very narrow range of values between 1 and 2 km. The ground instruments show a





more physical spread of the ash cloud locating it between 3 and 6 km. The comparison of the
ash plume height extracted from the IASI/MetopA UOXF iterative algorithm and the one
observed by the EARLINET network is shown in **Figure 3**b. It is evident from this figure that
the spread of plume heights found by the EARLINET network is higher than those found by the
Oxford iterative IASI algorithm leading to rather poor correlations. The estimate of the mean
is consistent between the datasets. This fact is demonstrated in the summary **Table *V*** which
gives the mean EARLINET and satellite ash plume height estimates. The large scatter bars
indicate the variability inherent in both sets of observations. We have to note here that the
UOXF-fast algorithm with fixed heights for the ash performs better for 600 hPa, which is
consistent with the average heights estimated by the nominal algorithm and the EARLINET
data, which range between 3 and 4 km. In all lidar-satellite comparisons there was no
indication that there were regions where the agreement between the two datasets is better,
due to their proximity to the source. However this conclusion is based, especially for certain
regions, on extremely few data.

## 3.2   COMPARISONS OF ASH OPTICAL DEPTH AND ASH LAYER HEIGHT WITH AIRBORNE LIDAR DATA

During May 2010 there were 12 flights of the UK's BAe-146-301 Atmospheric Research Aircraft
[Marenco et al., 2011], and during six of these volcanic ash was detected in the airborne lidar
measurements. In order to avoid contamination from cirrus clouds and mixed aerosol layers,
we only show comparisons with the satellite data for two flights, during which significant
levels of pure ash, not mixed with other aerosol types, were observed by the airborne lidar
measurements. The flight that took place on the 16[th] of May 2010 (see also Figure 1), started
at 12:55 U.T. and ended at 18:00 U.T. and the aircraft mostly flew over Scotland and northern
England. During this flight most of the ash was observed between 55° and 56°N.  The flight
that took place on the 17[th] of May over the Irish and North Sea, started a little earlier at 11:15
U.T. and ended at 16:58 U.T., and most of the ash was observed over the North Sea between
1° and 2°E.  As is demonstrated in ***Table III***, we only used spatial criteria to find coincidences
between the airborne lidar data and the satellite data of the same day, since both flights were
performed in the afternoon, while the satellite overpasses are close to 9:30 U.T. (GOME-2A
and IASI) and 21:30 U.T. (IASI only).  For GOME-2 we found coincidences only for the 17[th] of
May 2010.





In *Figure 4* we show the comparisons of the satellite ash optical depth and the airborne lidar
ash layer optical depth for 550 nm as a function of aircraft time (closest point in space). We
also show in the bottom row of *Figure 4* (4e and 4f) the flight track for the two flights
examined. On the path the actual flight time is indicated, in order to be able to identify the
spatial location that corresponds to the footprint of the lidar data. Since the time difference
between the flight measurement and the satellite overpass is large what we would actually
see from the comparisons is (a) if the aircraft and the satellite observe the plume over the
same area and (b) if they observe similar optical depth values. This would occur if the
dispersion, or transport, of the plume was not significant during the hours elapsing between
the satellite overpass and the aircraft measurement, within the spatial criteria we applied for
the comparisons. In the *Figure 4*a and 4b we show the comparisons between IASI ash optical
depth for the iterative and fast algorithm of UOXF versus the ash layer optical depth from the
airborne lidar measurements for the 16$^{th}$ of May 2010, where the measurements are shown
as function of time in U.T.C. In *Figure 4e* and *4f*, we plot the flight path for the two days (16
and 17 May 2010). Along the path the flight time in U.T.C is posted, while the different colours
along the flight path indicate the ash optical depth. As we can see, the satellite data processed
with the iterative UOXF algorithm captures the high AODs observed around 14:00 U.T. and
between 16:00 and 17:00 U.T. quite well, but fails to capture the peak observed between
15:00 and 16:00 U.T. In addition, it seems that the background is similar but that some larger
values are observed between the ash peaks. The situation is slightly different when examining
the comparisons between the aircraft data and the estimates from the UOXF fast algorithm
using a fixed height of the ash layer at 600 hPa.  In general, the UOXF fast algorithm estimates
smaller values (including the background); it captures well the peak observed around 14:00
U.T., overestimates the peak in AOD observed between 15:00 and 16:00 U.T. and it is hard to
tell if the smaller peak observed around 17:00 U.T. is well-depicted or not.
In *Figure 4c*, we present the comparisons between the aircraft data and the estimates from
the ULB-Eyja algorithm again for the 16$^{th}$ of May 2010. The satellite estimates follow quite
well all peaks observed in the aircraft data, however slightly misplaced. Checking the SEVIRI
ash imagery at http://fred.nilu.no for the 16$^{th}$ of May 2010 we observe an almost constant
west-east flow of dust throughout the day between 55$^{o}$N and 58$^{o}$ N, and thus this plume was
captured both by the morning and by the evening orbit of IASI, as well as by the aircraft when
flying over these latitudes between 14:00 and 16:00 UT. SEVIRI observed a plume after 17:00
UT south of 54$^{o}$ N moving southeast. The early evolution of this plume was captured by the
aircraft around 17:00 and its later evolution was captured over the same area by the evening



orbit of IASI. This plume evolution can partly explain the displacement observed, since the satellite data are not coincident in time with the aircraft data and the time in x-axis of the plots actually corresponds to different latitude/longitude of the comparisons.

In *Figure 4d* we present the corresponding comparisons between the aircraft data and the estimates from the GOME-2 KNMI algorithm for the 17th of May 2010, and in the right hand column of the last row of Figure 4 the corresponding flight path of the aircraft. The GOME-2 results capture the levels of the two AOD peaks observed in the aircraft measurements but fail to capture small scale variability in the AOD and the background levels. In these cases we actually compare only the morning orbit (9:30 UT) since GOME-2 is a UV/Vis sensor. SEVIRI images show a southeast movement of the ash plume starting east of the coast of England and going towards the Netherlands. The east-west motion of the aircraft over the sea captured this plume between 14:30 and 15:00, and GOME-2 observed this plume over the same area in the morning. Before 14:30 UT the aircraft was flying over land and did not observe any significant ash, so when compared with the morning observations of GOME-2 and considering the pixel size of GOME-2 and the collocation criteria applied, these measurements are actually compared with satellite data over the sea. Considering the large time difference between the flight and GOME-2 overpass and the much larger pixel size of GOME-2, compared to IASI, it is remarkable that the satellite data can quantitatively capture the ash optical depth in the greater flight area. **Table *VI*** summarizes the mean AODs values observed from the aircraft lidar and each of the satellite products examined.

Finally, in **Figure *5*** we present the comparisons of the ash layer height observed from the aircraft measurements and the corresponding effective ash height estimated from the UOXF-iterative algorithm based on IASI (Figure 5a) and the KNMI algorithm based on GOME-2 (Figure 5b). Considering the constraints induced by the collocation criteria, both algorithms show very good agreement with the corresponding heights estimated from the airborne lidar data in most of the collocations, with the ash height mainly ranging between 3 and 5 km. **Table *VII*** summarizes the mean ash layer height observed from the aircraft measurements and each satellite product examined.

## 4. SUMMARY AND CONCLUSIONS

The main aim of this work is to present the validation of improved and dedicated satellite-derived ash plume level assessments as part of the European Space Agency initiatives, in order to create an optimal "*End-to-End System for Volcanic Ash Plume Monitoring and Prediction systems*". The results shown are complementary to other satellite volcanic ash products, e.g.



from SEVIRI (Prata and Prata, 2012, Clarisse and Prata, 2015, WMO, 2015). Different aerosol
optical depth and ash plume height estimations from GOME2/MetopA and IASI/MetopA have
been assessed against collocated ground-based and airborne Lidar data for the 2010 eruptions
of the Icelandic volcano Eyjafjallajökull. The GOME2/MetopA measurements have been
analysed by the Royal Netherlands Meteorological Institute (KNMI) and the IASI/MetopA
observations by both the Université Libre de Bruxelles (ULB) and the University of Oxford
(UOXF). Different algorithm versions and parameters were examined and inter-compared.
Both aerosol optical depth and ash plume height satellite estimates were compared with
European Aerosol Research Lidar Network [EARLINET] lidar measurements and the UK's BAe-
146-301 Atmospheric Research Aircraft flying over the UK during the eruptive period.
▪  The KNMI GOME2 AOD over-estimates the ground-based values, showing quite high

12       values for cases where the LIDAR sees a low AOD. As a result, the *dust* algorithm shows

13       relatively low correlation coefficients of between 0.25 and 0.3 depending on the

14       spatiotemporal search radius, whereas the *Volz* algorithms perform slightly better, with

15       $r^2$ values ranging between 0.4 and 0.5. The KNMI/GOME2 data seem to suffer from the

16       spatial resolution of the satellite instrument which made the spatial criterion rather too

17       large hence precluding any conclusive comparisons when compared to the aircraft

18       measurements. The agreement between the satellite-derived and airborne lidar effective

19       ash heights differ only by 1 km on the average, indicating a homogenous spread of the

20       plume under the satellite's pixel. The KNMI GOME2 ash plume height comparisons are

21       not satisfactory, irrespective of the search radius, for either of the two algorithms. The

22       satellite ash height values seem to under-estimate the ground-based values, having a very

23       narrow range of values between 1 and 2 km and a mean of 2.07±1.22 km. In comparisons,

24       the ground instruments show a more natural spread between 3 and 6 km with a mean of

3.92±1.22 km. It is highly likely that the large GOME-2 pixel size smooths out any small

scale variability of the plume height, otherwise captured by the ground- based single point

measurements.

▪  The Oxford nominal IASI algorithm shows satisfactory AOD correlations against the

ground AODs, with coefficients ranging between 0.6 and 0.85, and, even though it

provides rather small optical depths, these are of the same order of magnitude as the

lidar. The algorithm presents quite good comparisons for the AOD patterns observed with

aircraft lidar. The Oxford nominal IASI algorithm ash plume height comparisons do not

show any significant correlation with the EARLINET estimates. The satellite estimates have

no spread in values compared to the lidar estimates, however both datasets show similar



average values, indicating that the satellite estimates can capture the average conditions. The results are better when compared with the aircraft lidar, where it seems that the satellite estimates follow the variability of ash height along the flight route; however they slightly underestimate the height values with a mean of 3.73±1.45km [compared to the aircraft mean of 4.30±2.00 km].

- ▪ The Oxford fast IASI algorithm also provides the same order of magnitude AOD estimates as the ground lidar, with the narrower spatio-temporal choice providing the most promising results: the 400 hPa product has a correlation of around 0.7 and the 800 hPa product a correlation of around 0.8. The Oxford fast IASI algorithm shows an excellent agreement with the aircraft lidar, where the 600 hPa product, that corresponds to the actual plume height, appears to perform best.

- ▪ The ULB AOD estimates are the most promising, showing the highest correlation coefficients, ranging between 0.74 and 0.91, depending on the spatio-temporal criterion chosen. This is also valid when we examine the ULB IASI – aircraft comparisons. The ULB IASI algorithm shows excellent agreement, both with respect to the absolute AOD values and with AOD features during the flight shown. The actual absolute AOD maxima are also represented best by this product.

Concluding, we note that, depending on the careful choice of collocation criteria, the satellite algorithms investigated here can observe the ash optical depth and plume height for large enough eruptions to a satisfactory degree. The results shown in this study are in line with the main finding of the dedicated WMO intercomparison study [2015] concerning the agreement between satellite ash products and validation data sets (for AOD correlations between 0.4 and 0.6 and ash layer height agreement within 2km) and in some cases the results shown here show better statistics. However, in order to quantify the levels of accuracy of the satellite assessments, eruptions with strong ash plumes need to be included in this type of validation exercise, since there were too few co-location scenes for most satellite products for the Eyjafjallajökull and Grimsvötn 2010 and 2011 eruptions, as examined in the course of the SACS/SMASH ESA projects. This validation study highlights the need for dedicated validation campaigns during volcanic eruptions. For future eruptions it could be recommended to fly instrumented aircraft along the satellite orbit in order to optimize the colocations between satellite data and aircraft-based observations. It is recognised that this would be a difficult campaign to plan, given that it is not possible to make precise long-term predictions of the eruptions.





## Acknowledgements

The comparison study was funded by the European Space Agency in the frame of the "Satellite
Monitoring of Ash and Sulphur dioxide for the mitigation of Aviation Hazards"-SACS-2 project.
The financial support for EARLINET in the ACTRIS Research Infrastructure Project by the
European Union's Horizon 2020 research and innovation program under grant agreement n.
654169 and previously under grant agreement n. 262254 in the 7thFramework Program
(FP7/2007-2013) is gratefully acknowledged. The UK's BAe-146-301 Atmospheric Research
Aircraft flown by Directflight Ltd and managed by the Facility for Airborne Atmospheric
Measurements (FAAM), which a joint entity of the Natural Environment Research Council
(NERC) and the Met Office. L.C. is a research associate with the Belgian F.R.S.-FNRS. LJV was
funded through the NERC National Centre for Earth Observation. RGG and EC were supported
by the NERC Centre for Observation and Modelling of Earthquakes, Volcanoes, and Tectonics
(COMET).





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



**Figure captions**
Figure 1. Characteristics of the FAAM flight of 16-5-2010. The flight track colored with AOD
(a), and the flight altitude versus time in UT along with a time-altitude cross section for the
aerosol extinction coefficient at 355nm (in $Mm^{-1}$) measured with the aircraft lidar (b).
Figure 2. Scatter plots between satellite ash optical depth at 550nm and EARLINET ash layer
optical depth at 532nm for GOME-2A (a) and (b), IASI-UOXF (c) and (d) and IASI-ULB (el)
products. Different colors correspond to different European domains. See Table I for more
details.
Figure 3. Scatter plots between satellite ash layer height and EARLINET ash layer height (in
km), for GOME-2A (a), and IASI-UOXF (b).
Figure 4.  Ash optical depth at 550nm and airborne lidar ash layer optical depth at 355 nm as
a function of aircraft time. IASI-UOXF products for the 16th of May 2010 (a( and (b), IASI-ULB
products for the 16th of May 2010 (c) and GOME-2A product for the 17th if May 2010 (d). The
flight tracks for these two days, colored with AOD are shown in (e) and (f)
Figure 5. Ash layer height and aircraft lidar ash layer height (in km) 355nm as a function of
aircraft time,.  GOME-2A for 17th of May 2010 (a), and IASI-UOXF for the 16th of May 2010 (b).





1  **Table I.** Locations of EARLINET lidar stations, their geographical coordinates and corresponding domain
2  assigned (C: Central Europe, N: North-Central Europe, SW: Iberian Peninsula, SE: Italy-Balkans).

| Site | Altitude a.s.l. (m) | Lat. (N) | Long. (E) | Domain |
|---|---|---|---|---|
| Andøya, Norway | 380 | 69.28 | 16.01 | N |
| Athens, Greece | 200 | 37.96 | 23.78 | SE |
| Barcelona, Spain | 115 | 41.39 | 2.11 | SW |
| Belsk, Poland | 180 | 51.84 | 20.79 | N |
| Bucharest-Magurele, Romania | 93 | 44.45 | 26.03 | SE |
| Cabauw, The Netherlands | 1 | 51.97 | 4.93 | N |
| Evora, Portugal | | | | SW |
| Garmisch-Partenkirchen, Germany | 730 | 47.48 | 11.06 | C |
| Granada, Spain | 680 | 37.16 | -3.61 | SW |
| Hamburg, Germany | 25 | 53.57 | 9.97 | N |
| Ispra, Italy | 209 | 45.82 | 8.63 | C |
| L'Aquila, Italy | 683 | 42.38 | 13.32 | SE |
| Lecce, Italy | 30 | 40.30 | 18.10 | SE |
| Leipzig, Germany | 100 | 51.35 | 12.44 | N |
| Linköping, Sweden | 80 | 58.39 | 15.57 | N |
| Madrid, Spain | 669 | 40.45 | -3.73 | SW |
| Maisach, Germany | 515 | 48.21 | 11.26 | C |
| Minsk, Belarus | 200 | 53.92 | 27.60 | N |
| Napoli, Italy | 118 | 40.84 | 14.18 | SE |
| Neuchâtel, Switzerland | 487 | 47.00 | 6.96 | C |
| OHP, France | 683 | 43.96 | 5.71 | SW |
| Palaiseau, France | 162 | 48.70 | 2.20 | N |
| Payerne, Switzerland | 456 | 46.81 | 6.94 | C |
| Potenza, Italy | 760 | 40.60 | 15.72 | SE |
| Sofia, Bulgaria | 550 | 42.67 | 23.33 | SE |
| Thessaloniki, Greece | 60 | 40.63 | 22.95 | SE |



1    **Table II.** Collocation criteria examined in the EARLINET-satellite comparisons

| Institute | Satellite product | Overpass time | Amount of Data In days | Co-location Criteria | Comments |
|---|---|---|---|---|---|
| KNMI | GOME2/MetopA | 09:30 LT | 14 | 3h & 300km<br>5h & 300km<br>3h & 500km<br>5h & 500km | |
| UOXF | IASI/MetopA-Nominal Algorithm | 09:30 LT<br>21:30 LT | 18 | 1h & 100km<br>3h & 300km | |
| UOXF | IASI/MetopA-Fast Algorithm | 09:30 LT<br>21:30 LT | 19 | 1h & 100km<br>1h & 300km<br>3h & 100km<br>3h & 300km | 3 fixed heights provided, 400 hPa, 600 hPa & 800 hPa |
| ULB | IASI/MetopA | 09:30 LT<br>21:30 LT | 48 | 1h & 100km<br>1h & 300km<br>3h & 100km<br>3h & 300km | |





1    **Table III.** Collocation criteria examined in the aircraft-satellite comparisons. The flights were
2    performed between 13:00 and 17:30 U.T..

| Institute | Satellite product | Overpass time | Amount of data in days | Co-location Criteria | Comments |
|---|---|---|---|---|---|
| | | | **Max # 5** | **No time constraint** | |
| KNMI | GOME2/MetopA | 09:30 LT | 1 | 100km/200km | |
| UOXF | IASI/MetopA-Nominal Algorithm | 09:30 LT 21:30 LT | 4 | 50/100/200km | |
| UOXF | IASI/MetopA-Fast Algorithm | 09:30 LT 21:30 LT | 4 | 50/100/200km | 3 fixed heights provided, 400, 600 & 800mbar |
| ULB | IASI/MetopA | 09:30 LT 21:30 LT | 5 | 50/100/200km | |





**Table IV.** Statistical mean values and associated standard deviation for the EARLINET and
the satellite ash optical depth estimates presented for collocated measurements.

| Product | Spatiotemporal criteria | EARLINET mean AOD at 532nm | Satellite mean AOD at 550nm |
|---|---|---|---|
| GOME-2A, KNMI *dust* | 300km & 5h | 0.19±0.22 | 1.29±0.48 |
| GOME-2A. KNMI *volz* | 300km & 5h | 0.19±0.22 | 1.32±0.69 |
| IASI, UOXF nominal | 100km & 1h | 0.12±0.12 | 0.08±0.08 |
| IASI, UOXF fast 400hPa | 100km & 1h | 0.12±0.12 | 0.10±0.04 |
| IASI, UOXF fast 600hPa | 100km & 1h | 0.12±0.12 | 0.17±0.12 |
| IASI, UOXF fast 800 hPa | 100km & 1h | 0.12±0.12 | 0.32±0.38 |
| IASI, ULB | 100km & 1h | 0.14±0.14 | 0.09±0.07 |

**Table V.** Statistical mean values and associated standard deviation for the EARLINET and the
satellite ash plume height estimates.

| Product | Spatiotemporal criteria | EARLINET mean and standard deviation [km] | Satellite mean and standard deviation [km] |
|---|---|---|---|
| IASI, UOXF nominal | 100km & 1h | 3.63±0.95 | 3.4±0.78 |
| GOME2/MetOp-A | 300km & 5h | 3.92±1.22 | 2.07±1.22 |



**Table VI.** Statistical mean values and associated standard deviation for the airborne lidar
and the satellite ash optical depth estimates at 550nm presented for collocated
measurements.

| Institute | Instrument & algorithm | Spatial criteria | Mean Satellite AOD levels | Mean Aircraft AOD Levels | Number of common observations |
|---|---|---|---|---|---|
| | | | | | |
| KNMI | GOME-2/MetOp-A | 200km | 0.42±0.03 | 0.23±0.15 | 64 |
| UOXF | IASI/MetopA Nominal Algorithm | 50km | 0.28±0.25 | 0.19±0.16 | 787 |
| UOXF | IASI/MetopA Fast Algorithm 400hPa | 50km | 0.20±0.30 | 0.19±0.16 | 776 |
| UOXF | IASI/MetopA Fast Algorithm 600hPa | 50km | 0.23±0.29 | 0.18±0.15 | 740 |
| UOXF | IASI/MetopA Fast Algorithm 800hPa | 50km | 0.30±0.40 | 0.18±0.16 | 732 |
| ULB | IASI/MetopA | 50km | 0.21±0.15 | 0.25±0.17 | 463 |

**Table VII.** Statistical mean values and associated standard deviation for the airborne lidar
and the satellite ash plume height estimates.

| Product | Spatial criteria | Aircraft mean and standard deviation [km] | Satellite mean and standard deviation [km] |
|---|---|---|---|
| IASI/MetOpA, UOXF nominal | 50km | 4.30±2.00 | 3.73±1.45 |
| GOME-2-MetOpA, KNMI | 200km | 3.87±1.70 | 5.62±0.54 |



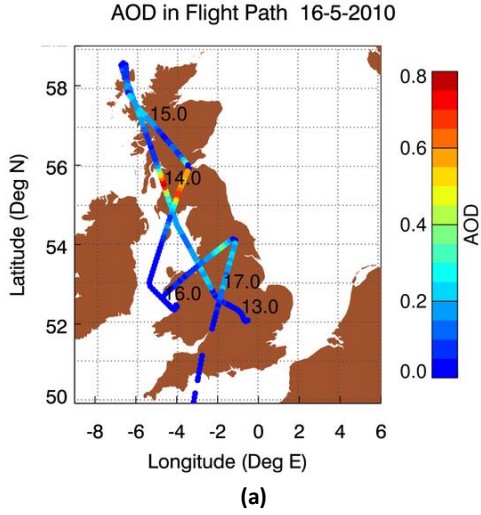

(a)

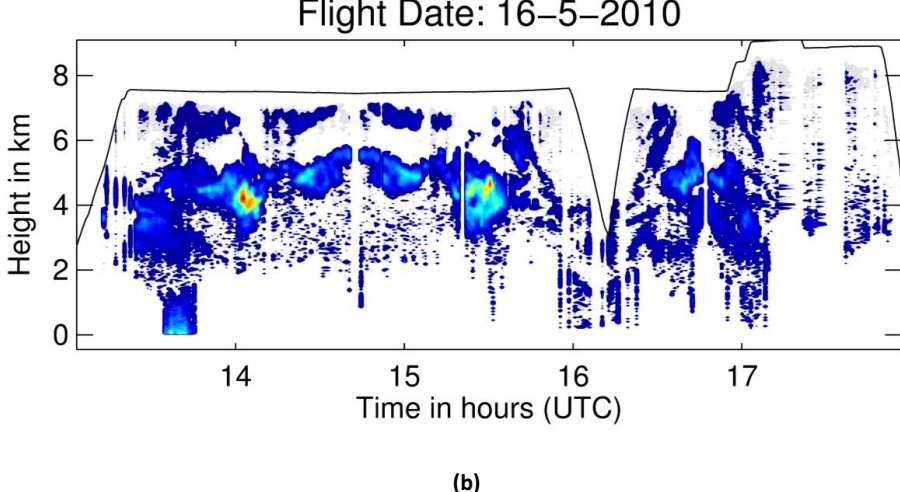

(b)

2      **Figure 1.**





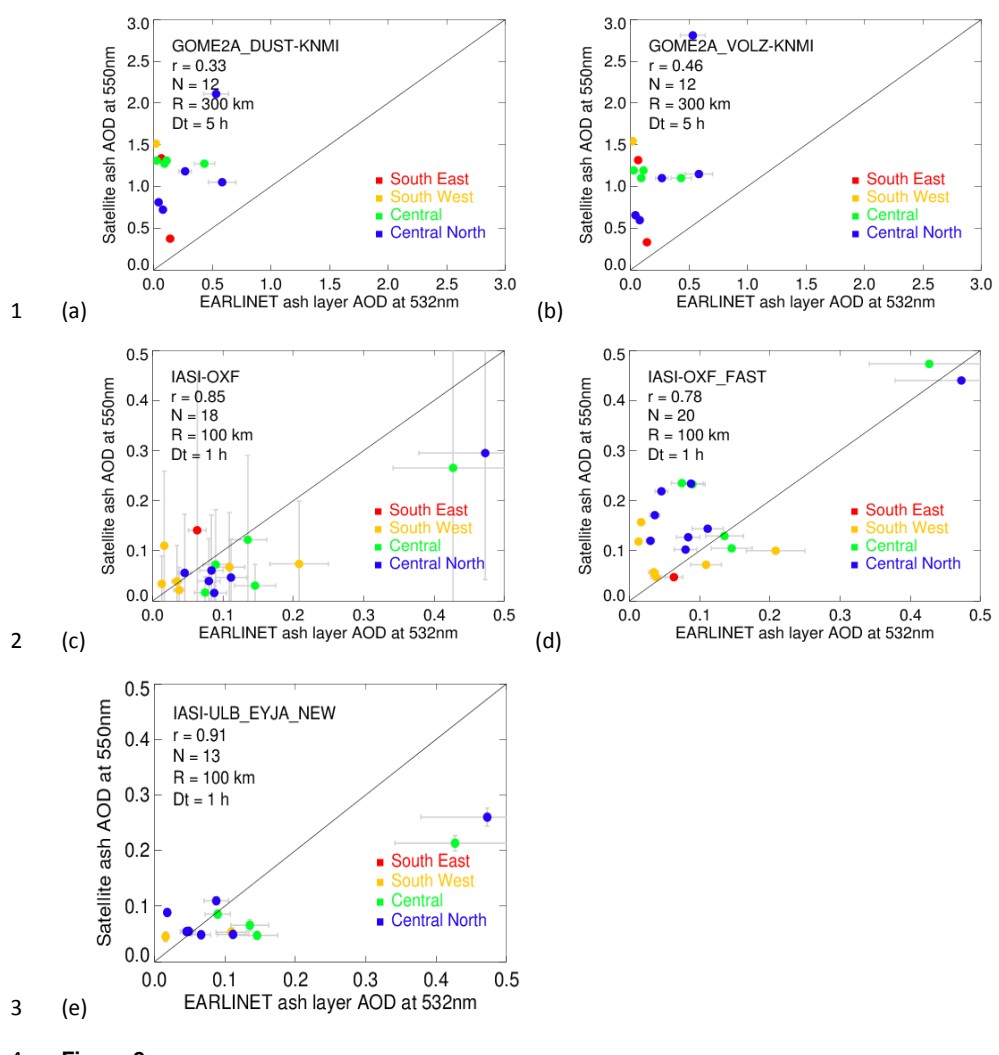

(a)
(c)
(e)
**Figure 2.**





2          (a)

3          (b)

4     **Figure 3.**




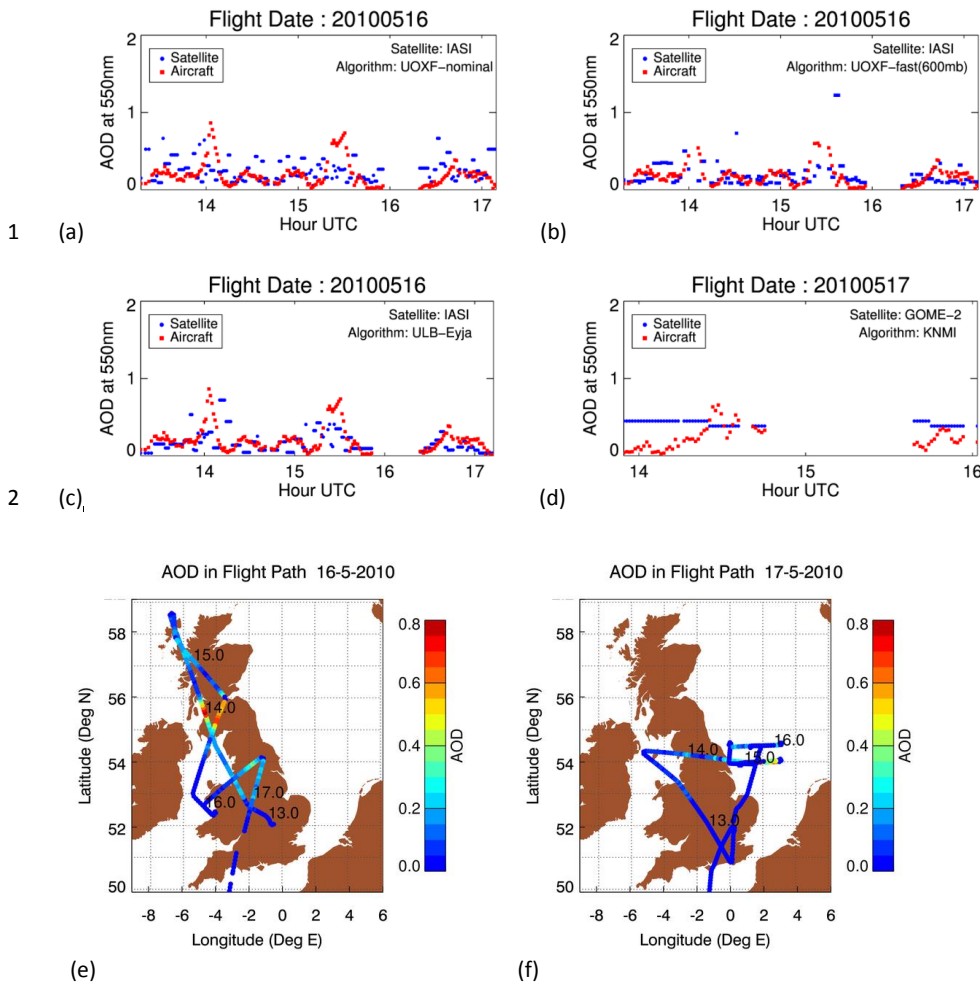

3    **Figure 4.**



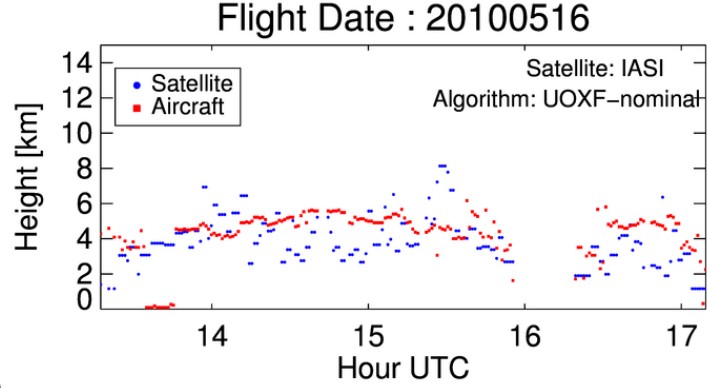

2        (a)

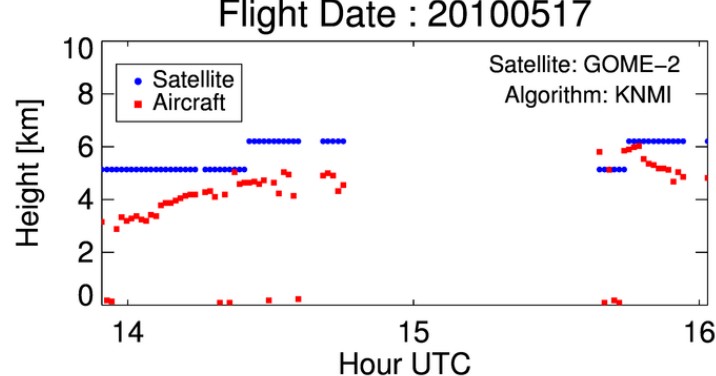

3        (b)

4    **Figure 5.**

