# Peer review of "Validation of ash optical depth and layer height retrieved from"

_Atmospheric Chemistry and Physics, 2015_

## Referee Comment (RC1) · Anonymous Referee #1 · 3 Feb 2016

This manuscript describes a the Evaluation of satellite observations of volcanic ash. I recommend publication in ACP after some minor revisions.

Comments:

At various places in the manuscript I would encourage the authors to openly discuss the very small sample sizes and that therefore the presented correlations are rather questionable. Everybody will acknowledge that (perhaps fortunately from the perspective of mankind) the set of available volcanic eruptions is very limited, so discussing this in more detail will do the manuscript no harm.
[Figure]

p.2 l. 20: replace "Final" by "The last"

p.2 l. 22: Potentially it would be advantageous to state that the classification of a "moderate" event is valid in terms of VEI, but not in terms of economic costs.

p. 4 l.1: Please introduce all abbreviations (SMASH).

p. 4 l.14: Is it possible to give either a citation for the AAI or to shortly describe the fundamental principles of ist derivation for those readers who are not familar with this method?

p. 4 l.17: In fact it is the absorbing AOD which is most dominant in AAI. A high AOD of a non-absorbing aerosol will not at all produce a high AAI. Consequently all AAI results are very sensitive to SSA.

p. 4 l. 27: What are the parameters of the log-normal distributions?

p. 4 l. 30: I would suggest to split the section 2.1.2 into two subsections, one for the UOXF algorithm and one for the ULB one.

p. 5 l. 2: What is the spectral resolution of the Eyja refractive indices? Is there already a publication on these?

p. 5 l. 3: The Pollack database includes pretabulated refractive indices as well as oscillator parameters for modelling these. The pretabulated indices have rather coarse spectral resolution (given the resolution of the IASI instrument). Which of the two sets has been used? And if it is the pretabulated ones, how has the interpolation to the required IASI channels been done?

p. 5 l. 8: What is the mode radius of the log-normal distributions?

p. 5 l. 20: Which meteorological input has been used for the RTTOV calculations? Moreover, to my understanding RTTOV is a radiative transfer model, which provides radiance or brightness temperatures or parameters like that which are simulated from given inputs including meteorology, AOD, PSD and such things. So these parameters

are input to RTTOV and not provided by the model. Or is there something like an "inverse mode" in RTTOV to obtain these parameters from the radiation field? Then RTTOV would be a suitable retrieval method and no further work would be required...

p. 7 l. 9: Please introduce all abbreviations (PCASP and CAS).

p. 8 l. 13: Would it be possible to provide the 532nm refractive index for the Eyjafjalla ash? Or is there already a publication on this?

p. 9 l. 19-20: When the authors say "dust" and "Volz", do they really mean two different algorithms, or do they rather mean two different complex refractive indices used as input for the same algorithm? Please clarify.

p. 9 l. 29: Give the FOV size of the IASI Instrument, not only thin clouds, but also partially cloudy observation could have an effect.

p. 10 l. 21: To which number does the number of coincidences decrease? Is the calculation of a correlation coefficient still useful then?

p. 15 l. 11: Please replace "excellent" by "very good".

p. 15. l. 17: The same - given the small sample size I would be rather shy about using the term "excellent".

Table II and Table III: Is "Amount of data in days" equivalent to "coincidences"? If not, please provide also the latter number.

---

## Referee Comment (RC2) · Anonymous Referee #2 · 9 Feb 2016

The paper presents the validation of ash optical depth and layer height as retrieved from satellites with ground bases and airborne based lidars. I recommend the publication with some minor revisions.

Some of my questions are already raised by RC 1 and thus I do not repeat them here.

General comment:

Correlation coefficient is not enough to define the correlation between satellite retrievals and ground based/airborne based measurements. Correlation coefficient is related with the linear regression between the two sets of data which does not follow

1:1 line. A high correlation coefficient alone does not mean that it exists a good fit between the data. An analysis of the residuals is required as well. Please consider a more complete analysis. Draw regression line along 1:1 line and discuss bias, residuals etc.

Specific comments:

pp 5 l 17: define TOVS

pp 7 l 7: please describe how LR was chosen and its implication on aerosol extinction coefficient

pp 7 l 23: what do you mean by "the closest point in space and time"? please provide numbers

pp 7 l 25: when talking about spatial and temporal filtering, do you refer to the lidar data? also, please describe the technical details of the filtering (e.g. moving average, resolutions etc)

pp 9 l 28-30: talking about cloud contamination in GOME-2A: isn't possible to screen the cloudy events?

pp 10 l 21: why the number of coincidences decreases?

pp 10 l 22-23: what is the physical meaning of the "ensemble average"(over the total number of coincidences) of AOD (table IV)? I mean relative error would have been useful.

pp 10 l 26: why do you mention the height of 800 hPa while Fig. 2 is based on the height of 600 hPa?

pp 11 l 8-9: the same question for the mean of ash plume height?

pp 12 l 2: what do you mean by "the closest point in space"? Between 50 km and 200 km as mentioned earlier?

pp 13 l 26: what do you mean by "very god agreement"? Please provide r2. Why didn't

you provide a scatter plot as in the case of Earlinet comparisons?

pp 13 l 32: Please rephrase "present the validation". As seen by these results, in my opinion, the validation is not satisfactory (based on present results). It is kind of an attempt to validate... How would you define the criteria for validation?

pp 23 l2-5: please reformulate. There is no middle panel in Fig 1.

pp 30 Fig. 2: middle plots: why there are 18 cases on the left plot and 20 cases on the right plot? Then the bottom plot has 13 cases? Please explain. I am also surprised by large r for the middle and lower plots. The data may be correlated but not with respect to 1:1 line. Please comment on this. The last plot in Fig. 2 looks to me very similar with the lower plot on Fig. 3 while they have quite different r. I know we talk about different quantities in the two figures but the points are spread quite similar.

pp 32 and pp 33: scatter plots as for Earlinet, including statistics (r, N) will help comparing the results and be consistent in validation criteria

---

## Author Comment (AC1) · 19 Apr 2016

Response to reviewer #1

We would like to thank the reviewer for his/her fruitful comments that helped to improve our manuscript.

"At various places in the manuscript I would encourage the authors to openly discuss the very small sample sizes and that therefore the presented correlations are rather questionable. Everybody will acknowledge that (perhaps fortunately from the perspective of mankind) the set of available volcanic eruptions is very limited, so discussing

this in more detail will do the manuscript no harm."

We certainly agree with the reviewer. The small sample issue has been already mentioned in the paper but is further emphasized in all relevant parts of the manuscript, especially in the conclusions.

"p.2 l. 20: replace "Final" by "The last"

The text has been modified accordingly in the revised manucript.

"p.2 l. 22: Potentially it would be advantageous to state that the classification of a "moderate" event is valid in terms of VEI, but not in terms of economic costs."

A relevant comment has been added in the introduction.

"p.4 l.1: Please introduce all abbreviations (SMASH)."

The SMASH abbreviation stands for "Satellite Monitoring of Ash and Sulphur dioxide for the mitigation of aviation Hazards" is explained in Page 4 and SACS-2 in page 3. In the revised manuscript will also be explained in the abstract.

"p. 4 l.14: Is it possible to give either a citation for the AAI or to shortly describe the fundamental principles of its derivation for those readers who are not familiar with this method?"

In the revised paper, two references have been included in the text directly after mentioning AAI. The revised text is: "The volcanic ash retrieval algorithm includes an estimation of the optical depth of an ash layer based on the Absorbing Aerosol Index (AAI) (Herman et al., 1997; Torres et al., 1998) as well as an estimation of the effective ash layer height"

"p. 4 l.17: In fact it is the absorbing AOD which is most dominant in AAI. A high AOD of a non-absorbing aerosol will not at all produce a high AAI. Consequently all AAI results are very sensitive to SSA."

We agree with the reviewer that the absorbing AOD is most dominant in AAI. The AOD from scattering aerosols would lower the AAI values. AAI results are very sensitive to SSA. In the text, we mentioned that AAI is sensitive to aerosol types, AOT …. The sensitivity of AAI to SSA is included implicitly in the sensitivity of AAI to the aerosol type.

"p. 4 l. 27: What are the parameters of the log-normal distributions?"

The parameters of the bi-mode log-normal size distribution for aerosols are effective radius, effective variance for fine and coarse modes, and the weight of the two modes. In our calculations, we used effective radius of 0.052 $\mu$m and effective variance of 1.697 $\mu$m for the fine mode, effective radius of 0.67 $\mu$m and effective variance of 1.806 $\mu$m for the coarse mode. The weight of the fine mode is 0.99565. This information is now included in the revised manuscript.

"p. 4 l. 30: I would suggest to split the section 2.1.2 into two subsections, one for the UOXF algorithm and one for the ULB one."

In the revised version we split section 2.1.2 in two subsections as suggested by the reviewer.

"p. 5 l. 2: What is the spectral resolution of the Eyja refractive indices? Is there already a publication on these?"

The spectral resolution of these indices is 1 cm-1. As far as we are aware of, these indices have not been published yet.

"p. 5 l. 3: The Pollack database includes pre-tabulated refractive indices as well as oscillator parameters for modelling these. The pre-tabulated indices have rather coarse spectral resolution (given the resolution of the IASI instrument). Which of the two sets has been used? And if it is the pre-tabulated ones, how has the interpolation to the required IASI channels been done?"

The pre-tabulated values have been used, interpolated using the Piecewise Cubic Hermite Interpolating Polynomial, which is shape preserving.

"p. 5 l. 8: What is the mode radius of the log-normal distributions?"

For the ULB the mode radius was retrieved along with the optical depth (see Moxnes et al, 2014). Mode radius is retrieved together with ash optical depth, plume altitude and surface temperature for the Oxford algorithm.

"p. 5 l. 20: Which meteorological input has been used for the RTTOV calculations? Moreover, to my understanding RTTOV is a radiative transfer model, which provides radiance or brightness temperatures or parameters like that which are simulated from given inputs including meteorology, AOD, PSD and such things. So these parameters are input to RTTOV and not provided by the model. Or is there something like an "inverse mode" in RTTOV to obtain these parameters from the radiation field? Then RTTOV would be a suitable retrieval method and no further work would be required."

ECMWF data are used as input to RTTOV. The Oxford iterative algorithm is a full optimal estimation retrieval scheme that calls iteratively the forward model. The forward model is based on RTTOV. RTTOV output for a clean atmosphere (containing gas but not cloud or aerosol/ash) is combined with an ash layer using the same scheme as for the Oxford-RAL Retrieval of Aerosol and Cloud (ORAC) algorithm (Thomas et al., 2009a, 2009b). In the text we have substituted the sentence: "RTTOV then provides probable values of AOD, effective radius and plume altitude [Ventress et al. 2015]." with: "The iterative retrieval scheme then provides probable values of AOD, effective radius and plume altitude [Ventress et al. 2016]."

"p. 7 l. 9: Please introduce all abbreviations (PCASP and CAS)."

PCASP stands for Passive Cavity Aerosol Spectrometer Probe and CAS stands for Cloud and Aerosol Spectrometer. Both have been introduced in the text.

"p. 8 l. 13: Would it be possible to provide the 532nm refractive index for the Eyjafjalla ash? Or is there already a publication on this?"

The refractive index used for Eyjafjalla ash is 1.572 +i 7.5e-06 at 530nm. This value is now mentioned in text of the revised manuscript.

"p. 9 l. 19-20: When the authors say "dust" and "Volz", do they really mean two different algorithms, or do they rather mean two different complex refractive indices used as input for the same algorithm? Please clarify."

We mean two different complex refractive indices used as input for the same algorithm. This is clarified in the revised manuscript.

"p. 9 l. 29: Give the FOV size of the IASI Instrument, not only thin clouds, but also partially cloudy observation could have an effect."

This is certainly the case for the ULB algorithm. For that reason, there is a strict quality check at the end of the algorithm, based on the retrieved parameters and the fit residual, which removes most of the cloudy observations. For the Oxford algorithm the spectral variability due to clouds is contained within the covariance matrix and, hence, if the cloud is below the ash plume, it should not present a problem. More details can be found in Ventress et al. 2016.

"p. 10 l. 21: To which number does the number of coincidences decrease? Is the calculation of a correlation coefficient still useful then?"

The number of coincidences with EARLINET stations are shown in the legends of Figure 2. For IASI-UOXF are around 18-20 and for IASI ULB are 13. In any case the sample is small for both data sets.

"p. 15 l. 11: Please replace "excellent" by "very good"."

The text has been modified accordingly

"p. 15. l. 17: The same - given the small sample size I would be rather shy about using the term "excellent"."

The text has been modified accordingly.

[Figure]

"Table II and Table III: Is "Amount of data in days" equivalent to "coincidences"? If not, please provide also the latter number."

No, it is not equivalent. With "amount of data" it is meant the number of days for which satellite retrievals were available. We introduced an additional column with the number of coincidences in the relevant tables. The number of coincidences are already shown in the legends of the plots.
* * *

---

## Author Comment (AC2) · 19 Apr 2016

Response to reviewer #2

We would like to thank the reviewer for his/her fruitful comments that helped to improve our manuscript.

"Correlation coefficient is not enough to define the correlation between satellite retrievals and ground based/airborne based measurements. Correlation coefficient is related with the linear regression between the two sets of data which does not follow 1:1 line. A high correlation coefficient alone does not mean that it exists a good fit between the data. An analysis of the residuals is required as well. Please consider a more complete analysis. Draw regression line along 1:1 line and discuss bias, residuals etc."

The reviewer is right. Additional results from the statistical analysis (mean bias, rms difference, the slope of the regression line) are shown in the tables and discussed. The figures have been modified accordingly in the revised manuscript.

"pp 5 l 17: define TOVS "

TOVS stands for TIROS Operational Vertical Sounder and has been introduced in the text.

"pp 7 l 7: please describe how LR was chosen and its implication on aerosol extinction coefficient"

An extinction-to-backscatter ratio (lidar ratio) of 60sr was used for the inversion of lidar signals; this lidar ratio was determined in such a way as to satisfy the constraints of a molecular signal below and above lofted layers. This has now been specified in the text in section 2.2.2. Please see Marenco et al, 2011 for details.

"pp 7 l 23: what do you mean by "the closest point in space and time"? Please provide numbers."

We consider for each coincidence the closest point in time and space within the colocation criteria shown in Tables II to VII. The colocation criteria define the upper limits, so the true coincidences in time and space are variable but within these limits. This has been explained in more detail in the text.

"pp 7 l 25: when talking about spatial and temporal filtering, do you refer to the lidar data? Also, please describe the technical details of the filtering (e.g. moving average, resolutions etc)"

What we mean here is that first the spatial colocation criteria have been applied to the satellite data and then the temporal ones. The sentence has been corrected accordingly.

"pp 9 l 28-30: talking about cloud contamination in GOME-2A: isn't possible to screen the cloudy events?"

What we mean here is that despite the screening of the cloudy events contamination could still be possible from thin clouds in the satellite retrievals considering the pixel size compared to the point lidar measurement. The text has been modified accordingly.

"pp 10 l 21: why the number of coincidences decreases? "

The ULB and UOXF algorithms have different criteria for considering a retrieval as successful.

"pp 10 l 22-23: what is the physical meaning of the "ensemble average"(over the total number of coincidences) of AOD (table IV)? I mean relative error would have been useful."

The tables have been modified to include additional statistics as mentioned already in our response to the 1st comment of the reviewer.

"pp 10 l 26: why do you mention the height of 800 hPa while Fig. 2 is based on the height of 600 hPa? "

Table IV shows results also from comparisons of the "fast algorithm" not shown in Figure 2 in order to reduce the number of subfigures. In figure 2 we only show the 600hPa because it shows the best overall agreement relative to the 800hPa and 400hPa estimates. A relevant comment has been added in the revised text.

"pp 11 l 8-9: the same question for the mean of ash plume height?"

The fast algorithm does not provide plume height, it assumes a fixed plume height.

"pp 12 l 2: what do you mean by "the closest point in space"? Between 50 km and 200 km as mentioned earlier?"

Yes, see our response to a previous comment on the colocation criteria.

"pp 13 l 26: what do you mean by "very good agreement"? Please provide r2. Why didn't you provide a scatter plot as in the case of EARLINET comparisons?"

The airborne lidar data give a time series of data for each measurement day. As data are not truly coincident (the overpass time being early in the morning and late in the evening whereas flights were near the middle of the day), volcanic plumes have undergone advection between the measurements. Looking at the data as a time series makes it easier to capture differences due to the misplacement of plumes. We do not show correlation coefficients and scatter plots for the aircraft-satellite comparisons because these are not truly coincident and thus the estimated statistics do not show a good correlation. This could be misleading concerning the usefulness of the comparisons and therefore we decided to show and discuss only qualitatively about the spatial consistency between the aircraft and the satellite data.

"pp 13 l 32: Please rephrase "present the validation". As seen by these results, in my opinion, the validation is not satisfactory (based on present results). It is kind of an attempt to validate... How would you define the criteria for validation?"

We modified the text as follows "present a first attempt to validate improved . . .". There were no predefined validation criteria, and no specifically designed validation campaign. Actually in this paper we examine the consistency between the lidar and the satellite data for volcanic ash retrievals. A relevant comment has been added.

"pp 23 l2-5: please reformulate. There is no middle panel in Fig 1."

The figure caption has been corrected accordingly.

"pp 30 Fig. 2: middle plots: why there are 18 cases on the left plot and 20 cases on the right plot? Then the bottom plot has 13 cases? Please explain. I am also surprised by large r for the middle and lower plots. The data may be correlated but not with respect to 1:1 line. Please comment on this. The last plot in Fig. 2 looks to me very similar with

the lower plot on Fig. 3 while they have quite different r. I know we talk about different quantities in the two figures but the points are spread quite similar."

The different number of cases depends on the number of successful satellite retrievals within the spatiotemporal criteria applied and is different for each algorithm. A relevant comment has been included in the text. The discussion on the correlation has been improved and the additional statistics have been included in the discussion.

"pp 32 and pp 33: scatter plots as for EARLINET, including statistics (r, N) will help comparing the results and be consistent in validation criteria"

See our response to a previous comment on aircraft-satellite comparisons.